# Water-Air Interface Imaging: Recovering the Images Distorted by Surface Waves via an Efficient Registration Algorithm

**DOI:** 10.3390/e24121765

**Published:** 2022-12-02

**Authors:** Bijian Jian, Chunbo Ma, Dejian Zhu, Qihong Huang, Jun Ao

**Affiliations:** 1School of Information and Communication, Guilin University of Electronic Technology, Guilin 541000, China; 2School of Artificial Intelligence, Hezhou University, Hezhou 542800, China

**Keywords:** image reconstruction, water–air imaging, image registration, patch search

## Abstract

Imaging through the wavy water–air interface is challenging since the random fluctuations of water will cause complex geometric distortion and motion blur in the images, seriously affecting the effective identification of the monitored object. Considering the problems of image recovery accuracy and computational efficiency, an efficient reconstruction scheme that combines lucky-patch search and image registration technologies was proposed in this paper. Firstly, a high-quality reference frame is rebuilt using a lucky-patch search strategy. Then an iterative registration algorithm is employed to remove severe geometric distortions by registering warped frames to the reference frame. During the registration process, we integrate JADE and LBFGS algorithms as an optimization strategy to expedite the control parameter optimization process. Finally, the registered frames are refined using PCA and the lucky-patch search algorithm to remove residual distortions and random noise. Experimental results demonstrate that the proposed method significantly outperforms the state-of-the-art methods in terms of sharpness and contrast.

## 1. Introduction

Imaging through water is an important research direction of underwater optics and marine optics, which is widely employed in environmental, economic and military application fields [1,2,3,4,5,6]. Unlike other underwater imaging systems, the main challenge in this imaging scenario comes from the air–water interface. Random fluctuations of surface waves will introduce a variety of effects, such as non-single viewpoint distortion [7], double images [8], scene distortion, illumination caustics [9] and black holes [10,11]. These effects can cause extreme distortion and motion blur in the images. Therefore, inverting these distortions and retrieving high-quality images remains a formidable task.

Several efforts have been made in earlier works to address the issue of retrieving the distortion-free image from a video sequence distorted by surface waves. If the turbulence of surface waves is small, averaging the sequence or the blind deconvolution method [12] will perform well, whereas the reconstruction result using them will be extremely blurry once the waves are strong. In prior research, better techniques were to select the image patches with the least distortions and combine them [13,14,15,16,17,18]. The limit of these methods lies in the fact that they assume the corresponding patches from different times are spatially invariant, which causes such algorithms to handle only slightly fluctuating situations and is easy to cause information loss. Later, another distortion mitigation approach, an image restoration scheme of water surface modeling based on image sequence is described in [19,20,21,22,23,24]. However, it is challenging to portray different complicated fluid motion events in real environments using such a simplistic spatial model.

Recently, registration technology, which was initially developed to recover atmospheric turbulence images, has been applied to this problem. The typical strategy [25,26,27] is to use the temporal mean as the reference frame and to improve mean–frame correlation by blurring the sharp frames with a blur kernel. The mean and frames are then iteratively refined via robust registration. Although this scheme can significantly eliminate severe geometric distortions, it is prone to inadvertent misalignment and motion blur. Once in a strong wave environment, such effects will be more pronounced. Subsequently, Zhen Zhang et al. [28,29] attempt to reconstruct a higher-quality reference frame by using a lucky-patch search and a deblurring process. they attempted to guide the registration of a sharper boundary to remove geometry distortions. However, they ignore two key factors that affect the registration process: (1) The premise of a lucky patches search is that the corresponding regions from different frames are spatially invariant or approximate; otherwise, it will result in the apparition of double images in the synthetic frame; (2) The deconvolution procedure required in deblurring is often ill-posed, even the most advanced deblurring algorithms, such as [30,31,32], frequently fail and may result in the appearance of visual artifacts (commonly described as “artificial” edges) [33]. Although the contrast of the image is improved in their processing results, the overall performance evaluation is not satisfactory, and even inferior to Oreifej’s method [25] in individual indicators.

In this work, we present a video-sequence-based method for image reconstruction. At first, a new lucky-patch search method is used to reconstruct a high-quality reference frame, where all warped frames are aligned in the first iteration, followed by the patch search in subsequent iterations. Secondly, the warped frames are then registered against the reference frame using image registration to eliminate the severe geometry distortions in the images. Finally, the registered frames are refined using PCA and the lucky-patch search algorithm to remove residual distortions and random noise.It is worth noting that in the registration stage, we introduce a hybrid optimization strategy to speed up the optimization process of the control parameters. The experimental results show that the proposed approach outperforms the state-of-the-art methods in terms of sharpness and contrast.

## 2. Related Work

Researchers from various fields have attempted to tackle the problem of recovering the real scene from a distorted video sequence. According to Cox–Munk law [34], the normals of the water surface generally follow a Gaussian distribution when the interface is sufficiently large and calm. As such, numerous approaches related to the lucky-patch were presented Efors et al. [13] described the reconstruction issue as a manifold embedding problem in which a patch with the smallest global distance was chosen as the distortion-free patch. Donate et al. [14] presented a multi-stage clustering approach to eliminate regions with significant amounts of translation and motion blur, independently. Wen et al. [15] used a bi-spectral analysis approach to transform the reconstruction issue into a phase recovery issue. A.V. Kanaev et al. [16] developed a method for image restoration based on optical flow and the lucky region. The lucky patch was determined using an image evaluation metric and the current frame’s nonlinear gain coefficient at each location. Later in [17], A.V. Kanaev et al. propose a structural tensor-oriented image quality measure (STOIQ) to evaluate image quality. Zhang et al. [18] recently proposed “center evolution imaging”, a simple yet effective progressive recovery method that improves restoration quality by constantly updating following warped frames. Subsequently, several researchers attempted to recover the distortion-free image based on model methods. Tian et al. [19] used a model-based tracking approach to reconstruct the real scene and instantaneous shape of the water surface simultaneously. In later work, they proposed a data-driven iterative algorithm [20] and introduced the operation of “pulling back”. In Ref [21], Seemakurthy K et al. proposed methods to remove motion blur caused by unidirectional cyclic waves and circular ripples, respectively. More recently, Li et al. [22] demonstrated their trained convolutional neural network that can remove dynamic refraction using a single image. James et al. [26] came up with a method for restoring underwater images using both compressive sensing (CS) and a local polynomial image representation. Thapa et al. [24] addressed a distortion-guided network (DG-Net) to restore distortion-free underwater images. They first use a physically constrained convolutional network to estimate the distortion map from the refracted image. Then, using the distortion map as a guide, a generative adversarial network was used to retrieve a sharp image with no distortion.

Recently, another typical approach, registration technology, was used to tackle this issue. Oreifej et al. [25] proposed a two-stage restoration method that first removed structured noise using a robust registration process, and then eliminated sparse noise by rank minimization. Hu Wenrui et al. [26] proposed a motion field kernel regression-based method to reconstruct distortion-free images. Halder et al. [27] introduced a pixel-shift map-based restoration method. First, the method uses image registration to estimate pixel displacement maps of warped images, and then reconstructs distortion-free scenes based on these maps. Zhen Zhang et al. [28,29] attempted to reconstruct a reference frame using a lucky-patch search and a deblurring process. They tried to guide the distorted frame registration to sharper boundaries to remove geometric distortions, but the results were not convincing. Cai et al. [35] proposed a reconstruction algorithm for recovering distorted images based on a two-round registration algorithm, which obviously improves the restoration accuracy but reduces the computational efficiency of image registration.

With the aim to raise the restoration accuracy, visual effects, and computational efficiency, we propose a new image reconstruction approach that utilizes a high-quality reference image to guide the distorted frames toward focusing on the sharper boundary or region. Moreover, an optimization strategy which combines the JADE with LBFGS methods is introduced to accelerate the iterative registration process. Unlike other methods, ours does not need a particular camera height [19], image priori [20], or special illumination [36,37]. Our approach is similar to the state-of-the-art [25,28,29,30] in that it only needs a short sequence (61 frames) rather than 800 frames in [14], 300k frames in [22], 100 frames in [23], and 43,600 frames in [24], where it is the prerequisite for underwater imaging systems to achieve covert observation.

## 3. Methods

Assume a water-deformed video sequence $V=\{I_{1},I_{2},\cdots ,I_{n}\}$ in which each frame $I_{k}\in \;R^{w\times h},\;k=1,\cdots ,n$ is warped by an unknown deformation Γ(x). Our primary objective is to rebuild a distortion-free frame sequence $V_{f}=\{I_{f1},\cdots ,I_{fn}\}$ and a high-quality dewarped frame $I_{F}$. The framework of the image reconstruction method in this paper is shown in Figure 1.

### 3.1. Principles

As Figure 1 shows, we optimize both the reference frame and registered frames using a robust iteration registration procedure, in which at each iteration the distorted frames are registered against the reference frame which is closer to the real-scene image. At first, a new lucky-patch search strategy is proposed to generate a higher-quality reference frame than the mean, in which we align all distorted frames to the average frame using an initial registration in the first iteration, the purpose of which is to remove the spatial variability between images introduced by random fluctuations of the water surface. We then perform the patch search algorithm in subsequent iterations. Next, similar to [25], we estimate the blur kernel to make sure that the distorted frames have the same blur level as the reference frame. Subsequently, a non-rigid image registration based on the hybrid optimization strategy is employed to estimate the motion vector field Γ(x) and remove the serious geometric distortions. The hybrid optimization strategy consists of a global optimization algorithm JADE [38] and a local optimization algorithm LBFGS [39], with the purpose of avoiding deformation parameters falling into local optimum and improve the registration process.

After a few iterations, the robust iteration registration will converge to new registered frames $V_{r}=\{I_{r1},I_{r2},\cdots ,I_{rn}\}$ which are less distorted and blurred. However, there are still several sparse random noise and misalignments among them, which are difficult to remove by robust registration. In order to acquire a distortion-free sequence $V_{f}=\{I_{f1},\cdots ,I_{fn}\}$ and a high-quality dewarped frame $I_{F}$, the registered frames are refined using PCA technology and the lucky-patch search method described above to remove residual distortions and random noise.

### 3.2. Lucky-Patch Search with Initial Registration

In order to reconstruct a higher-quality synthetic image than the temporal mean of the sequence, we introduce an image-quality-assessment-driven lucky-patch search approach. Firstly, the video frames are divided into N overlapping sub-regions with the same size, $R_{k}=\left\{P_{1},P_{2},\cdots ,P_{N}\right\},\;k=1,\cdots ,n\;$, where the overlapping area of adjacent image patches is 50%. The k-Medoids clustering algorithm [41] is then used to try to search for the set of high-quality patches with the fewest distortions from each patch sequence $P_{m}=\left\{b_{m1},\cdots ,b_{mn}\right\}\;,m=1,\cdots ,N$. Subsequently, all of the high-quality patches from each patch sequence are averaged into a single patch. Finally, a complete fused image is constructed by splicing all the average patches together.

In this paper, we use edge strength similarity (ESSIM) [42] as a clustering criterion. ESSIM is a structural similarity index which has better performance than other image quality metrics such as SSIM [43] and FSIM [44]. EESIM is used for measuring the similarity between the patch $b_{mk}\;,\;m=1,\cdots ,N\;;\;k=1,\cdots ,n$ and the mean $u_{m}$ of the corresponding patch sequence. EESIM is given by
(1)$$\mathrm{ESSIM}\left(u_{m},b_{mk}\right)=\frac{1}{M}\sum_{\eta =1}^{M}\frac{2E\left(u_{m},\eta \right)E\left(b_{mk},\eta \right)+c}{E{\left(u_{m},\eta \right)}^{2}+E{\left(b_{mk},\eta \right)}^{2}+c}$$
where M denotes the total number of pixels. $E\left(u_{m},\eta \right)$ and $E\left(b_{mk},\eta \right)$ represent the total edge-strength and the edge-strength of $b_{mk}$ at the η th pixel, respectively. The c is the scaling parameter. 

Furthermore, in order to ensure the corresponding patches from different times are spatially invariant or approximate, we first register all frames to the mean at the 1th iteration. The framework to reconstruct the reference frame is shown in Figure 2.

### 3.3. Image Registration Based on the JADE & LBFGS Optimization Strategy

Similar to [30], a nonrigid image registration algorithm based on the JADE and LBFGS optimization strategies is applied to estimate the deformation Γ(x) and remove the distortion. Assume the image domain as $\Omega =\left\{(x,y)\;|\;0\leq x\;<\;X,\;0\leq y<Y\right\}$, the goal of the iterative registration is to find the optimal transformation Γ(x) to guide registering the video sequence to the reference frame. The deformation Γ(x) can be expressed as
(2)$$\Gamma (x)=\Gamma_{\mathrm{local}}(\Gamma_{\mathrm{global}}(x))$$
where $\Gamma_{\mathrm{global}}$ and $\Gamma_{\mathrm{local}}$ are the global transformation and the local transformation, respectively. $\Gamma_{\mathrm{global}}$ describes the overall motion of the moving image, which is generally represented by an affine transformation with six parameters, namely
(3)$$\Gamma_{\mathrm{global}}(x\;;\;\Theta )=\left[\begin{array}{cc} \theta_{11} & \theta_{12}\\ \theta_{21} & \theta_{22} \end{array}\right]\left[\begin{array}{c} x\\ y \end{array}\right]+\left[\begin{array}{c} \theta_{13}\\ \theta_{23} \end{array}\right]$$
where $\Theta =\left\{\theta_{11},\theta_{12},\theta_{13},\theta_{21},\theta_{22},\theta_{23}\right\}$ represents the set of the six parameters.$\Gamma_{\mathrm{local}}$ models the local deformation of the distorted frames. Local deformation is difficult to characterize using parameterized transformations since the wavy water surface introduces anisoplanatism effects for imaging through water. Instead, we employ the B-spline based free-form deformation model (FFD) to represent the local deformation. The basic ideal of FFD is to describe the geometry distortions between the moving image and the reference frame by manipulating an underlying mesh of control points. Let Φ as a $n_{x}\times n_{y}$ mesh of control points $\varphi_{i,j}$ with uniform spacing, the FFD can be expressed as
(4)$$\Gamma_{\mathrm{local}}\left(x\;;\;\Phi \right)=\sum_{t=0}^{3}\sum_{l=0}^{3}B_{t}(u)B_{l}(v)\varphi_{i+t,j+l}$$
where $i=\left\lfloor x/s_{x}\right\rfloor -1,\;j=\left\lfloor y/s_{y}\right\rfloor -1,\;u=x/s_{x}-\left\lfloor x/s_{x}\right\rfloor ,\;v=y/s_{y}-\left\lfloor y/s_{y}\right\rfloor$ and $B_{l}$ is a standard B-spline basic function which defines the weight of the control points contributing in the deformation of the arbitrary point x in the image. The B-spline basic functions are given by
(5)$$\left\{\begin{array}{l} B_{0}(v)={\left(1-v\right)}^{3}/6\\ B_{1}(v)=(3v^{3}-6v^{2}+4)/6\\ B_{2}(v)=(-3v^{3}+3v^{2}+3v+1)/6\\ B_{3}(v)=v^{3}/6 \end{array}\right.$$

We minimize a cost function associated with the global transformation parameters Θ and the local control parameters Φ to determine an optimal transformation Γ(x). The function can be defined as
(6)$$E\left(\Theta ,\Phi \right)=C_{\mathrm{similarity}}\left(\Theta ,\Phi \right)+\lambda C_{\mathrm{smooth}}\left(\Phi \right)$$
where $C_{\mathrm{similarity}}\left(\Theta ,\Phi \right)$ is the similarity metric function which describes the difference between the reference frame $I_{R}$ and the distorted frame $I_{k}$. Here, we utilize the squared sum of intensity differences (SSD) to denote the similarity metric function, namely
(7)$$C_{\mathrm{similarity}}\left(\Theta ,\Phi \right)=\frac{\sum_{x\in \Omega }\left[I_{k}\left(\Gamma \left(x\right)\right)-I_{R}\left(x\right)\right]}{w\times h}$$
where $I_{k}(\Gamma (x))$ and $I_{R}(x)$ are the intensity of the corresponding pixels of two images, respectively. Furthermore, λ is the weighting parameter which defines the tradeoff between the alignment of the two images and the smoothness of the transformation. $C_{\mathrm{smooth}}(\Phi )$ is a regularization penalty term that controls the smoothness of the transformation. The penalty term in 2-D is expressed as follows:(8)$$C_{\mathrm{smooth}}\left(\Phi \right)=\frac{1}{A}\int_{0}^{X}\int_{0}^{Y}\left[{\left(\frac{\partial^{2}\Phi }{\partial x^{2}}\right)}^{2}+{\left(\frac{\partial^{2}\Phi }{\partial y^{2}}\right)}^{2}+2{\left(\frac{\partial^{2}\Phi }{\partial x\partial y}\right)}^{2}\right]\;dxdy$$
where A denotes the image domain.

Unlike the previous works [25,26,27,28,29,30], we no longer use the LBFGS search algorithm or the gradient descent method alone to minimize Equation (6), but combine the JADE and LBFGS algorithms as the search strategy that is described in detail in Section 3.3.1.

#### 3.3.1. Optimization Strategy

Consider the minimization of a smooth nonlinear function f: ℝτ→ℝ,
(9)minfδ
where δ denotes the multi-variable vector, δ∈ℝτ, and τ is the length of variables.

First of all, first-order techniques, such as gradient descent, can be utilized to solve Equation (9). However, gradient descent may not perform well in ill-conditioned situations [45], and the convergence speed of the method will be slower as it approaches the objective function’s lowest point. Typically, second-order techniques, such as Newton’s method, are used to address this deficiency. Unfortunately, Newton’s method needs the inverse of the Hessian matrix $\nabla^{2}f(\delta )$ at each iteration, and its computing cost is prohibitively expensive for variables with a large dimension. In quasi-Newton approaches such as [39,46,47], an approximated Hessian matrix is used to take advantage of the curvature while also reducing computing complexity. Generally speaking, BFGS performs the best in practice. The limited-memory variant of the original BFGS algorithm (LBFGS) reduces the computational cost of each iteration even more, making it one of the most effective algorithms in this domain. However, similar to other local optimization methods, the LBFGS algorithm easily converges to the local optimal solution once the initial value $\delta_{0}$ is not selected properly. Therefore, how to select the initialization parameters becomes the key to obtaining the optimal parameters.

JADE [38] is a simple and effective global optimization method, which is a variant of the differential evolution algorithm. Differential evolution follows the general process of evolutionary algorithms. The initial population $\left\{\delta_{z}^{0}=(\delta_{z,1}^{0},\delta_{z,2}^{0}\cdots ,\delta_{z,\tau }^{0})|\;z=1,2,\cdots ,NP\right\}$ is randomly generated according to a uniform distribution, where NP is the population size. If nothing is known about the solution of the objective function, the initial population is randomly selected. If a preliminary solution is available, the initial population is usually generated by adding random deviations from the normal distribution. The JADE algorithm then enters a cycle of evolutionary operations: mutation, crossover, and selection. After repeating the above process, the algorithm will gradually approach the global optimal solution.

Therefore, in order to avoid the transformation parameters (Θ,Φ) falling into a local optimum, we attempt to minimize Equation (6) by utilizing JADE with LBFGS algorithms. Among them, JADE is used to calculate a neighboring solution to the global optimal solution, and LBFGS is used to obtain the optimal solution. Furthermore, for computational efficiency, we do not use an iterative multi-resolution search strategy [40,41] any longer in the stage where the affine transformation parameters Θ are optimized. In practice, it is sufficient to utilize the combination optimization strategy. 

Specifically, the optimization process in this paper can be divided into the following two stages. In the first stage, the integrated JADE and LBFGS algorithms are used to optimize the global transformation parameters Θ. In the second stage, the local transformation parameters Φ are optimized through a hierarchical multi-resolution strategy [40] such that the resolution of the control mesh is increased, along with the image resolution in a coarse-to-fine fashion.

### 3.4. Post-Processing

After a few iterations, we can obtain a new image sequence $V_{r}=\{I_{r1},I_{r2},\cdots ,I_{rn}\}$ which is less distorted and blurred. However, there are still several sparse random noise and misalignments among them, which are difficult to remove by robust registration. In order to obtain a distortion-free image sequence $V_{f}=\{I_{f1},\cdots ,I_{fn}\}$, we refine the image sequence using PCA technology [48]. Subsequently, we perform the patch search strategy described in Section 3.2 again to select the patches with less distortion from the registered frames to fuse a higher-quality single image $I_{F}$. Unlike the first fusion, we first divide each patch sequence into five groups rather than two groups using the k-Medoids clustering algorithm. The reason for this is that the differences between adjacent images are already small after iterative robust registration and sparse denoising. Moreover, to improve the registration efficiency, we can add a suitable termination condition ξ(v) to determine if the registration process has ended or not. The comprehensive process for image reconstruction is described in Algorithm 1.

**Algorithm 1:** The correct format is:An efficient method for recovering the images distorted by surface waves **Input:** Distorted Video sequence  V={I1,I2,⋯,In} , Ik∈ℝw×h,k=1,2,⋯,n **Output: Distortion-free sequence and high-quality frame**  Vf={If1,⋯,Ifn} , Ifk∈ℝw×h,k=1,2,⋯,n ; IF∈ℝw×h **While**
 r≤maxiteration L
**do**
**if**

r=1



$I_{R}⇐TemporalMean(V);$


**else**


$I_{R}⇐Patches\_search(V);$


**end**

**for**

k=1:n

  B⇐EestimateBlurKernle(Γ);

$I_{blur}⇐GaussianFilter(I_{k},B);$



$\Gamma ⇐ComputerDeformation(I_{blur},I_{R});$



$V_{r}(:,:,k)⇐Dwrap\left(I_{k},\Gamma \right);$


**end**


V={I1,I2,⋯,In}⇐Vr={Ir1,Ir2,⋯,Irn},Irk∈ℝw×h,k=1,2,⋯,n;



r=r+1;

 **end** $V_{f}=\{I_{f1},\cdots ,I_{fn}\}\;⇐PCA\left(V_{r}\right)\;,r=L$Computer the frame $I_{F}$**for** each patch sequence $P_{m}=\left\{b_{m1},\cdots ,b_{mn}\right\}\;,m=1,\cdots ,N$

$Q=\left\{q_{1},\cdots ,q_{n}\right\}⇐ComputeESSIM(P_{m})$



$P_{best}=\{b_{m1},b_{m2},\cdots ,b_{mc}\}⇐KmedoidCluster(P_{m},Q),\;1\leq c\leq n$



$u_{m}⇐TemporalMean(P_{best})$


**end**


$U=\left\{u_{1},u_{2},\cdots u_{m}\right\},m=1,2,\cdots ,N$



$I_{F}⇐\;Fusion(U)$



## 4. Results

In the experiment, the proposed method described above was implemented on MATLAB (Math Work Co., Natick, MA, USA). To validate the effectiveness of our method, we first conducted extensive experiments using several standard underwater sequences from [19]. Then, we further tested the proposed method with two typical real-scene sequences from [18]. Moreover, we made a comparison between our experimental results with other state-of-the-art methods, such as Tian’s method [19], Oreifej’s method [25], Zhen Zhang’s method [28] and Tao Sun’s method [29]. The maximum number of iterations for all image registration algorithms was set as five and each patch of the corresponding sequence was set in the same size. The source codes of our method are available online in [49].

### 4.1. Test with the Data Set from the Air-to-Water Imaging Scenario

To more accurately validate the proposed method, the experiment employed the same standard underwater scene sequences from [19]. The data sets contain five image sequences (as Table 1 shows) of which each sequence contains 61 frames. Table 1 shows the size of each frame and patch utilized in the patch search procedure.

In order to quantify the results of these methods, we employed three standard image quality/similarity metrics for quantitative evaluation, namely PSNR [30], MSE [6] and SSIM. Above metrics are classified into full-reference that represent the different between the moving frame and the reference frame. The expressions for MSE, PSNR, and SSIM can be described as
(10)$$\mathrm{MSE}=\frac{1}{w\times h}\sum_{x=1}^{w}\sum_{y=1}^{h}{\left[F(x,y)-G(x,y)\right]}^{2}$$
(11)$${\mathrm{PSNR}=10\log}_{10}(\frac{\max{\left(G\right)}^{2}}{\frac{1}{w\times h}\sum_{x=1}^{w}\sum_{y=1}^{h}{\left[F(x,y)-G(x,y)\right]}^{2}}$$
(12)$$\mathrm{SSIM}=\frac{\left(2u_{F}u_{G}+c_{1})(2\sigma_{FG}+c_{2}\right)}{\left(u_{F}^{2}+u_{G}^{2}+c_{1})(\sigma_{F}^{2}+\sigma_{G}^{2}+c_{2}\right)}$$
where F and G are the estimated image and the reference image, respectively. max(·) represents the maximum value. $u_{F}$ and $u_{G}$ are the averages of F and G, respectively. $\sigma_{F}$ and $\sigma_{G}$ denote the variance of F and G, respectively. $\sigma_{FG}$ is the covariance of F and G. $c_{1}=(\;0.01\times \beta \;)$, $c_{2}=(\;0.03\times \beta \;)$ denote the constants, in which β represents the dynamic range of pixel values. Moreover, since the template image for “Checkerboard” and “Large fonts” are unknown, a novel non-reference underwater image quality measurement method UIQM [50] is used as another quality metric. The UIQM metric is given by
(13)$$\mathrm{UIQM}=\lambda_{1}\times \mathrm{UICM}+\lambda_{2}\times \mathrm{UISM}+\lambda_{3}\times \mathrm{UIConM}$$
where UICM, UISM and UIConM represent the image colorfulness metric, the image sharpness metric and the image contrast metric, respectively. $\lambda_{1},\lambda_{2},\lambda_{3}$ are constants which are generally set as 0.0282, 0.2953 and 3.5753. Due to the image sequences being gray images, the UICM is identically equal to zero.

#### 4.1.1. Analysis of Restoration Results

Figure 3 shows the recovery results of our method and other state-of-the-art methods. The image quality evaluation results are shown in Table 2. Compared with Oreifej’s results, our results perform better in term of sharpness and contrast and preserve more detail of the underwater scene, which illustrate that the higher-quality reference frame than the mean better guides the warped image to focus on sharp boundaries or areas. In addition, we find that the results from Z. Zhang and T. Sun’s methods still contain lots of residual distortions such as misalignment, ghosting and local deformation, etc. The main reason is that they ignore the premise of spatial invariance, which is very important for the lucky-patch search strategies. Unlike the methods [28,29], we attempted to reconstruct a high-quality reference frame utilizing a new patches search strategy which added an alignment process at the first iteration. The results show that our method can reconstruct all of the underwater scenes and make high-quality results that are better than others.

#### 4.1.2. Analysis of Computational Efficiency

In term of computational efficiency, we compared the running time of different algorithms using two different termination conditions. In order to ensure fairness, we performed these methods on the same laptop computer. The operating system contains a CPU (i7-6700HQ) and RAM (8 GB). The data set was “Middle fonts”. Moreover, in Oreifej’s work [25], it was proved that the processing time of Tian’s method was twice that of Oreifej’s method, so it is no longer compared with Tian’s method here in terms of computational efficiency.

Firstly, the termination condition is set as the maximum number of iterations. During the registration process, the comparison of the running times of different algorithms is shown in Figure 4. The average iterative times of different methods are 261.6 s, 236.4 s, 245 s, and 187.6 s, respectively. Because the length of the data set is 61, the average cost times of each frame are 4.29 s, 3.88 s, 4.02 s and 3.07 s, respectively. The results demonstrate that the proposed method may expedite image registration and shorten image processing time.

For the purpose of raising efficiency, we also consider another approach, which is to set the iteration termination condition as the difference between the registered frames and the average of the current sequence V. Similar to [25], the termination condition can be expressed as
(14)$$Τ\left(V\right)=\frac{\sum_{k}\sum_{x}\left|I_{k}(x)-I_{mean}(x)\right|}{w\times h\times n}$$
where the threshold is set to 0.025. 

The restoration time of different methods is shown in Figure 5. We can notice that our method’s overall running time and iteration numbers are significantly less than those of other approaches. For Oreifej’s results, the main reason is due to the severely blurred mean image. The blurring of the mean will become worse once the fluctuation of the water surface becomes stronger. In order to remove geometry distortions, the video frames are then also blurred to the same fuzzy level. However, the above strategy may reduce the registration efficiency due to the large amount of overlap and ghosting present in the mean image. As for Zhen Zhang’s and Tao Sun’s methods, the main reason is due to two aspects: (1) they ignored the effect of spatial inconsistency between different frames on the fused image. (2) They did not consider the problem of artificial artifacts introduced by deblurring methods. These problems can easily lead to distorted images being registered to the wrong sharp edges, which often have priority in their strategy. Thereby, a new patches search strategy is applied to generate the reference frame in our method, where we first perform an alignment operation for the input video sequence. Our reference frame is better than the mean image in terms of sharpness and contrast. Moreover, another important reason is that our combinatorial optimization strategy is significantly better than other algorithms in term of computational efficiency, which we discuss in detail in Section 5.3.

Therefore, the method has better computational efficiency when the number of iterations is the same or unknown, which helps to promote the application of this technique in underwater covert observations, such as a virtual periscope.

### 4.2. Test with the Data Set from the Water-to-Air Imaging Scenario

In order to further confirm the effectiveness of the proposed approach,, we tested data sets from another imaging scenario, the underwater-to-air imaging scenario. The data sets come from Zhang’s experiment [18], whose source data are also available online. The data sets contain two typical sequences, “gravity” and “ripple”, in which the first sequence is distorted by a gravity wave and the other is distorted by wind-generated capillary waves. Each sequence is also composed of 61 frames. The maximum number of iterations for all image registration algorithms was set to five, and each patch in the related sequence was set to the same size.

The recovery results with different methods are shown in Figure 6. Moreover, we compared these algorithms with a non-reference quality evaluation metric, UIQM, considering that template images are not available for the above datasets. The comparison results of the UIQM metric for different methods are shown in Table 3. The results show that our algorithm is still effective in water-to-air imaging scenarios and produces higher-quality results.

## 5. Discussion

### 5.1. Analysis of the Lucky-Path Search Strategy

#### 5.1.1. Selection the Image Quality Metric

During the patch fusion process, ESSIM was chosen as the quality metric. We also tested our algorithm using SSIM, FSIM, and NIQE [51] metrics, where NIQE is a non-reference metric. The fusion results based on the different image quality metrics at the 2th iteration stage are shown in Figure 7. In addition, the quality scores for the above four quality indicators (here, SSIM is used as the evaluation standard) are 0.6500, 0.6571, 0.6510, and 0.6289, and the computational efficiency is 4.56 s, 11.03 s, 78.12 s, and 33.03 s, respectively. According to the above results, considering the computational efficiency of the algorithm, we chose EESIM as the image quality metric at the patch fusion stage. The results show that other reliability metrics can be considered in future work to further optimize our patch fusion process.

#### 5.1.2. Analysis of the Evolution of Reference Frame

Figure 8 demonstrates the evolution of the reference frame for the different methods in the registration process. The image quality evaluation results for the reference frames of different algorithms for each iteration are shown in Table 4 In Z. Zhang and T. Sun’s approaches [28,29], they directly perform the patch search process to generate reference frames, ignoring the issue of spatial differences in input sequences. For this reason, the reference frames inevitably produce ghosting, misalignment, and overlapping effects as shown in Figure 8 (remarked by the red rectangle). Furthermore, we find that there are artifacts in the reference frame that are not related to image details in the results of Z. Zhang’s method, which are clearly the result of blind convolution. In other words, deblurring operations similar to blind convolution algorithms are obviously infeasible and tend to degrade the restoration results. However, our method can always generate higher-quality reference frames than the mean, which would guide the registration process and lead to better restoration results, as shown in Figure 3. The reference frames generated by our patch search strategy perform better in sharpness and contrast than other methods. As shown in Figure 8, compared with other methods, the text information in the reference frame reconstructed by our method is clearer and more accurate (remarked by the green rectangle). The results show that it is essential to align the input sequence before applying the lucky block fusion technique. Obviously, our patch search strategy always produces higher-quality results.

### 5.2. Analysis on the Choice of Reference Frame Deblurring or Frame Blurring

Figure 9 demonstrates the evolution process of the synthetic frame $I_{F}$ for the “Middle fonts” sequence in the iterative registration process with three cases. As shown in Figure 9, our method recovers high-quality results after several iterations under all three strategies. However, textual details in images, such as “Imaging”, “Tracking”, and “computer”, can only be clearly reconstructed using our method.

Furthermore, it is worth noting that, compared to the first strategy, both frame blurring and reference frame deblurring strategies can produce sharper results, which seems to mean that we can try to seek a more effective deblurring scheme to obtain high-quality results.

### 5.3. Comparison of the Optimization Strategy

The comparison results of the running time of different optimization strategies after each iteration are shown in Figure 10. The results show that our optimization strategy outperforms the state-of-the-art optimization methods used in [25,26,27,28,29,30,41] in terms of computational efficiency. The main reasons can be attributed to the following two aspects: (1) The combined strategy of JADE and LBFGS is used to estimate the initial value of the optimization parameters (Θ,Φ), which avoids the local optimum problem. (2) In the affine parameter optimization stage, the multi-resolution optimization strategy is no longer used.

### 5.4. Compared with Deep Learning Method 

To more thoroughly validate its performance, we compare our method to another cutting-edge strategy, deep learning techniques, which Tian’s method [19] also belongs to. In terms of repeatability, James’ technique [23], the source code for which is available online, was chosen as a reference method. The proposed method and James’ method were tested using identical testing data, “large fonts” and “ripple.” Figure 11 shows the restoration effects of the two methods. The results show that these two methods can effectively reduce the geometric distortion and motion blur of the image and restore better results. However, in terms of contrast and sharpness, our algorithm shows better performance. As Figure 11 shows, when the water surface disturbance is small, both James’ method and our method can restore the distorted scene image well, and our algorithm shows better sharpness (as indicated by the green markers). Once the water surface disturbance becomes stronger, the James method will not be able to recover high-quality results, especially the details of the scene (marked by the red rectangle). However, our method can reconstruct better boundaries and more complete details. Experimental results show that simple spatial distortion models similar to [23] may fail in heavily disturbed environments. However, our algorithm still produces high-quality results.

## 6. Conclusions

This paper proposes a new image reconstruction scheme with distorted video sequences while keeping image restoration accuracy and computational efficiency in mind. The main contributions are as follows: (1) The optimal combination of lucky- patch fusion and iterative registration algorithms is realized, which effectively eliminates the ghosting problem in a fused reference frame and raises the recovery accuracy of the distorted frames; (2) Introduce a mixed optimization strategy of JADE and LBFGS to accelerate the registration process.

We extensively experimented on the proposed method using five standard underwater sequences from [18] and the two typical data sets from [19]. Experiments show that our approach performs better in terms of sharpness and contrast. Therefore, this means that our algorithm has better development prospects in applications such as underwater or airborne objective monitoring, obstacle avoidance for underwater vehicles, seafloor mapping, etc.

To further raise the recovery accuracy and computational efficiency, we will develop a more effective method to generate a higher-quality reference frame in the future. Moreover, it is worth noting that the machine learning methods [53,54,55,56,57,58] have unique advantages in improving image recognition. For application scenarios, where covert monitoring is not considered, once the model is established, it can be used as a means of quickly detecting objects.

## Figures and Tables

**Figure 1 entropy-24-01765-f001:**
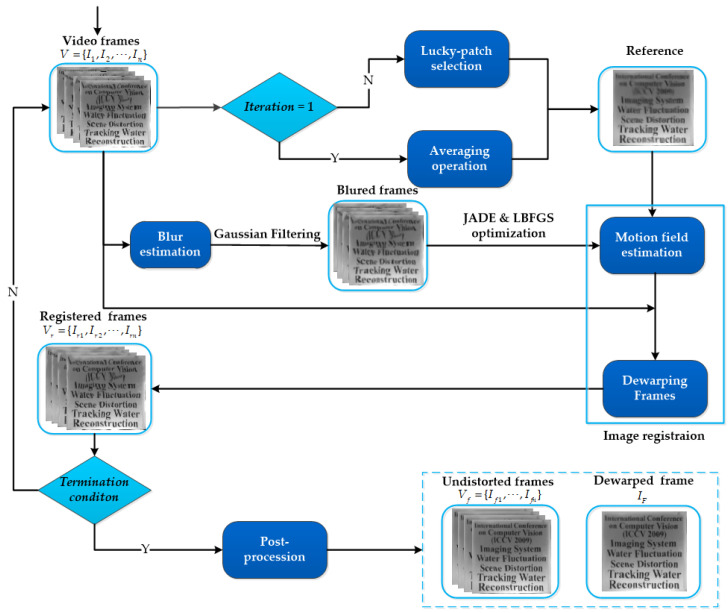
The framework of the image reconstruction method in this paper.

**Figure 2 entropy-24-01765-f002:**
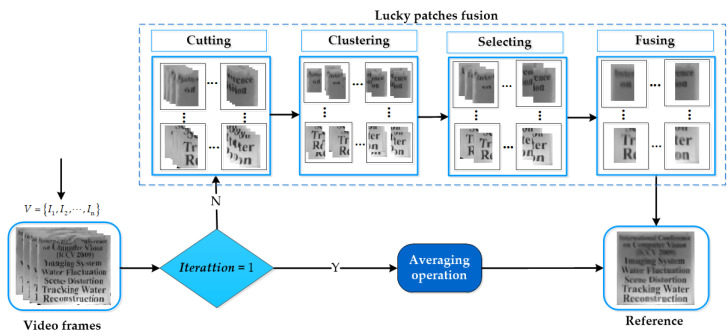
The framework to reconstruct high-quality reference frame.

**Figure 3 entropy-24-01765-f003:**
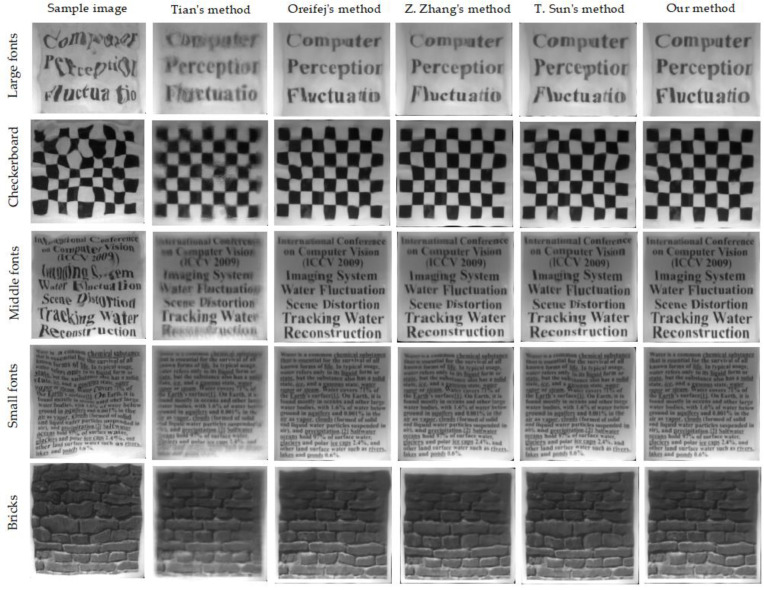
The recovery results of different methods. Left to right: the sample image, the results of Tian’s method [19], the results of Oreifej’s method [25], the results of Z. Zhang’ method [28], the results of T. Sun’ method [29] and the result of our method.

**Figure 4 entropy-24-01765-f004:**
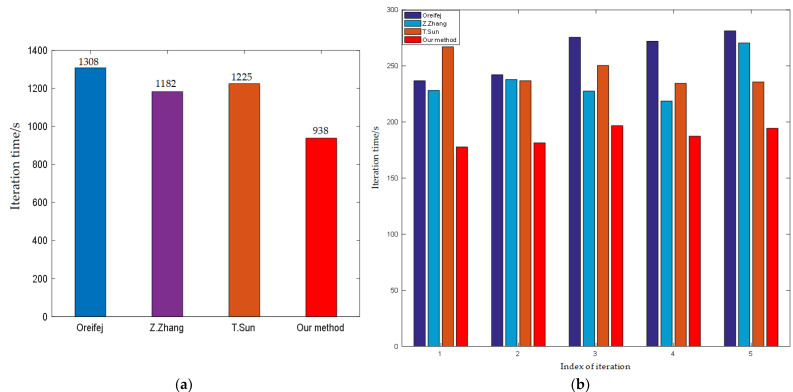
The running time of different methods after 5 iterations. (**a**) A comparison of total running time of different methods; (**b**) A comparison of running time of different methods after each iteration.

**Figure 5 entropy-24-01765-f005:**
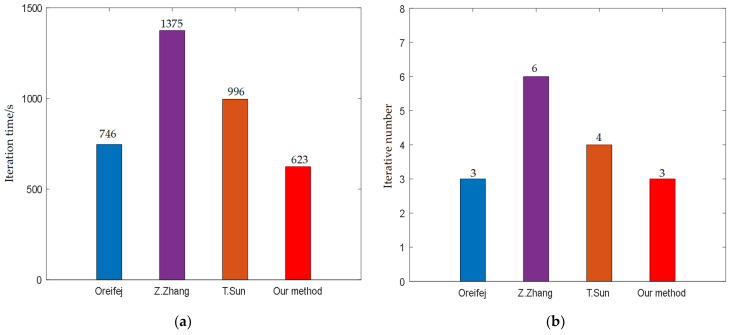
The running times of different methods at the same precision threshold. (**a**) The total running time of each method when the criterion for termination is satisfied; (**b**) The number of iterations for each method when the criterion for termination is satisfied.

**Figure 6 entropy-24-01765-f006:**
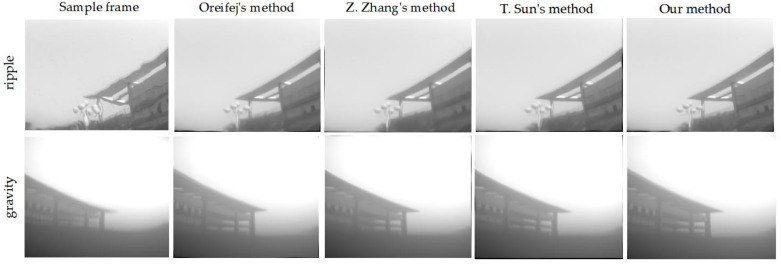
The restoration results with different methods in water-to-air imaging scenarios.

**Figure 7 entropy-24-01765-f007:**
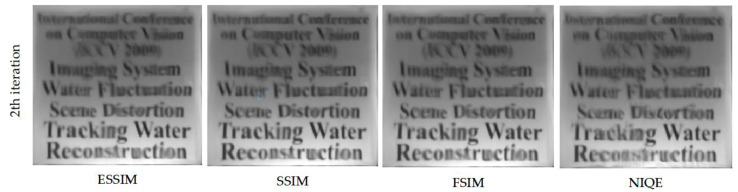
The fused frame based on the different image quality metrics at the 2th iteration stage. From left to right: ESSIM, SSIM, FSIM and NIQE.

**Figure 8 entropy-24-01765-f008:**
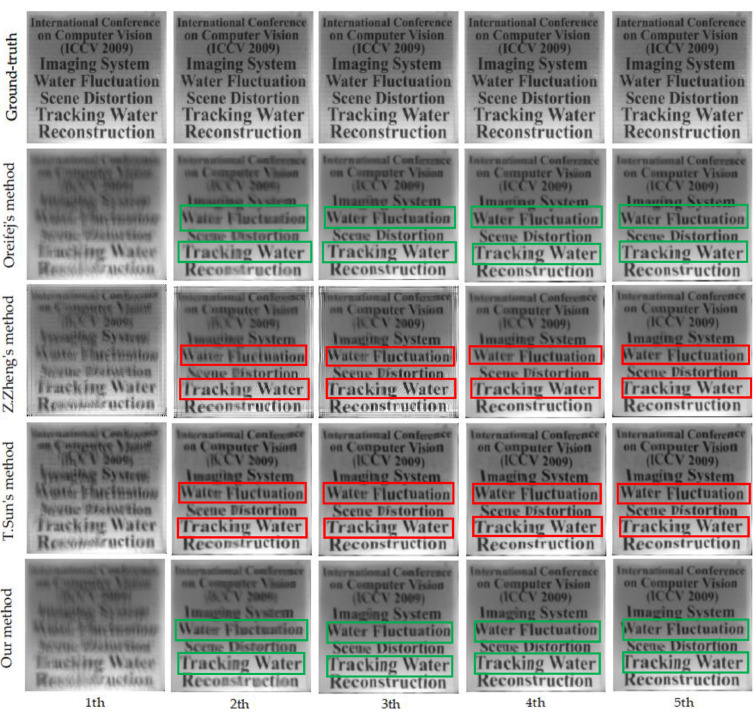
Evolution of the reference frame of the different methods in registration process. Top to bottom: the ground-truth image, the result of Oreifej’s method [25], the results of Z. Zhang’s method [28], the results of T. Sun’s method [29] and the results of our method.

**Figure 9 entropy-24-01765-f009:**
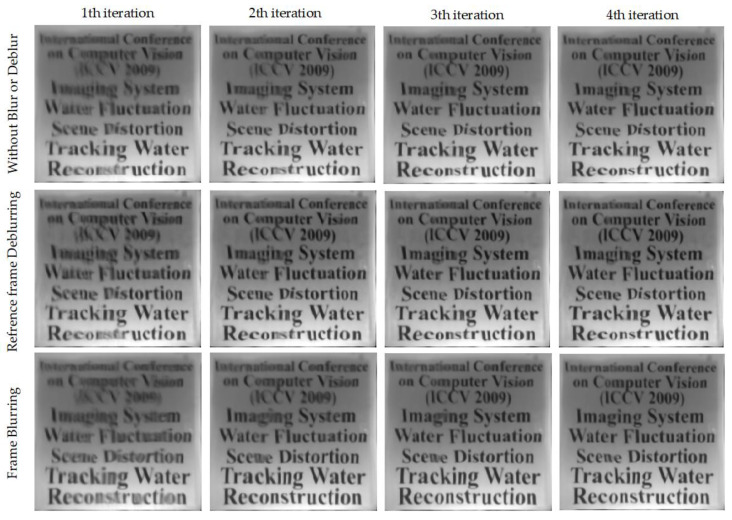
Evolution of the synthetic frame $I_{F}$ in registration process. Top to bottom: Our method employed without blurring or deblurring, with reference frame deblurring using [52], and with frame blurring using our method.

**Figure 10 entropy-24-01765-f010:**
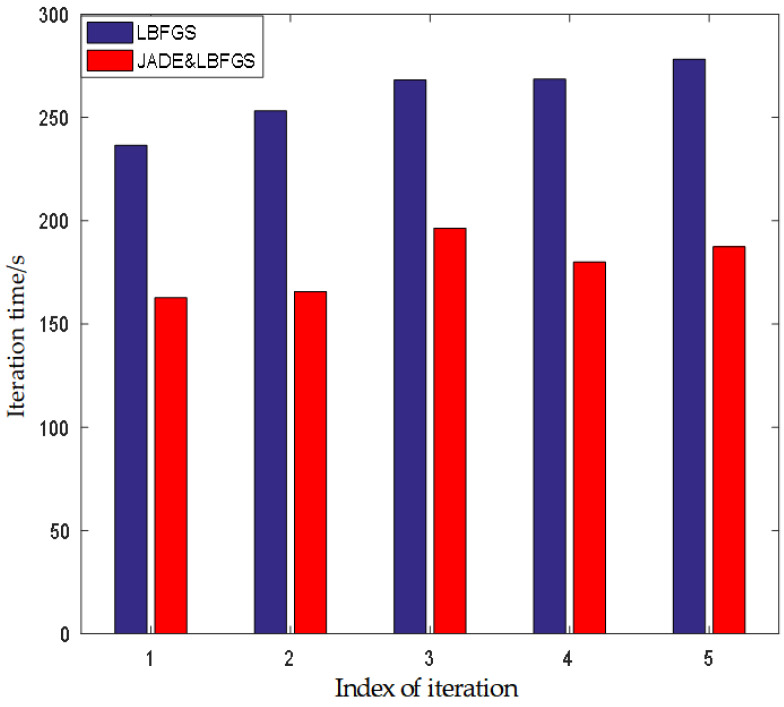
The comparison of running time of different optimization strategy after each iteration.

**Figure 11 entropy-24-01765-f011:**
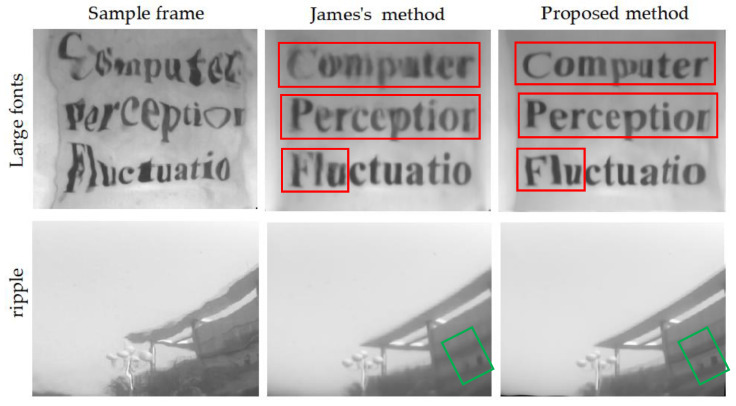
The comparison between James’s method [23] and the proposed method. Left to right: the sample images, the results of James’s [23], and the results of our method.

**Table 1 entropy-24-01765-t001:** The frame size and patch size of the different video sequences.

	Brick	Checkerboard	Large Fonts	Middle Fonts	Small Fonts
frame size	268×292	238×285	238×324	253×293	255×284
patch size	76×116	68×114	68×108	84×58	102×94

**Table 2 entropy-24-01765-t002:** Numerical comparison based on the standard data sets ^1^.

Data Sets	Methods	MSE(L)	PSNR(H)	SSIM(H)	UIQM(H)
Middle fonts	Tian [19]	0.0079	21.0275	0.6598	2.5215
	Oreifej [25]	0.0073	21.3829	0.7412	2.6182
	Z. Zhang [28]	0.0164	17.8407	0.5773	2.5773
	T. Sun [29]	0.0119	19.2361	0.6137	2.6277
	Our method	0.0053	22.7591	0.7891	2.7056
Small fonts	Tian [19]	0.0043	23.6916	0.6475	2.5852
	Oreifej [25]	0.0038	24.1490	0.7098	2.8036
	Z. Zhang [28]	0.0061	22.1768	0.5689	2.7409
	T. Sun [29]	0.0068	21.7069	0.5424	2.7853
	Our method	0.0036	24.4752	0.7243	2.8409
Bricks	Tian [19]	0.0108	19.6800	0.6017	2.4071
	Oreifej [25]	0.0051	22.9278	0.6361	2.5631
	Z. Zhang [28]	0.0182	17.3951	0.4112	2.4807
	T. Sun [29]	0.0118	19.2700	0.4373	2.5846
	Our method	0.0051	22.9278	0.6497	2.7126
Large fonts	Tian [19]	×	×	×	2.2026
	Oreifej [25]	×	×	×	2.0533
	Z. Zhang [28]	×	×	×	2.1072
	T. Sun [29]	×	×	×	2.1218
	Our method	×	×	×	2.2851
Checkerboard	Tian [19]	×	×	×	2.4701
	Oreifej [25]	×	×	×	2.4900
	Z. Zhang [28]	×	×	×	2.5504
	T. Sun [29]	×	×	×	2.5240
	Our method	×	×	×	2.5928

^1^ Because the template images of “Large ” and “Checkerboard” are unavailable, the PSNR, MSE, SSIM are presented with “×

**Table 3 entropy-24-01765-t003:** Numerical comparison based on Zhang’s data sets [18] (Bold values are the best).

Data Sets	UIQM(H) Oreifej/Z. Zhang/T. Sun/Our method
ripple	1.9231/1.9544/2.0128/**2.1573**
gravity	1.3119/1.3218/1.3266/**1.3347**

**Table 4 entropy-24-01765-t004:** Image quality evaluation results of reference frames for different methods.

Iteration	Methods	MSE(L)	PSNR(H)	SSIM(H)
1th	Oreifej	0.0120	19.1934	0.4847
	Z. Zhang	0.0149	18.2634	0.3940
	T. Sun	0.0117	19.3080	0.4907
	Our method	0.0120	19.1934	0.4847
2th	Oreifej	0.0107	19.7244	0.5576
	Z. Zhang	0.0171	17.6689	0.4293
	T. Sun	0.0122	19.1431	0.5904
	Our method	0.0072	21.4257	0.6491
3th	Oreifej	0.0073	21.3932	0.7123
	Z. Zhang	0.0225	16.4742	0.4364
	T. Sun’s	0.0130	18.8677	0.6054
	Our method	0.0057	22.4323	0.7475
4th	Oreifej	0.0072	21.4247	0.7340
	Z. Zhang	0.0121	19.1718	0.6115
	T. Sun	0.0132	18.7993	0.6086
	Our method	0.0056	22.5441	0.7700
5th	Oreifej	0.0072	21.4247	0.7407
	Z. Zhang	0.0172	17.6536	0.5737
	T. Sun	0.0133	18.7726	0.6090
	Our method	0.0055	22.5986	0.7781

## Data Availability

All data or codes used to support the findings of this study are available from Ref [49].

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
