# Peer review of "Water-Air Interface Imaging: Recovering the Images Distorted by Surface Waves via an Efficient Registration Algorithm"

_entropy, 2022, doi:10.3390/e24121765_

Round 1

Reviewer 1 Report

Comments and Suggestions for Authors:

Imaging through the wavy water-air interface is challenging since the random fluctuations of water will cause complex geometric distortion and motion blur in the image, seriously affecting the effective identification of the monitored object. With the aim to raise the restoration accuracy, visual effects and computational efficiency, the authors proposed a new image reconstruction approach that utilizes a higher quality reference image to guider the distorted frames toward focus on the sharper boundary or region. Moreover, they integrated the JADE and LBFGS algorithms as an optimization strategy to expedite the iterative registration process. Finally, they performed extensive comparisons of their results with other state-of-the-art methods, validating the effectiveness of the algorithm. Regarding the conception of the paper and the presented results, the paper can be graded as good, but requires some improvements and correction before publishing.

To publish this paper, the following revisions are recommended to improve the quality of this paper:

(1)Please briefly explain the concepts of patch search strategy and lucky block fusion, and point out their differences。

(2)Abstract: The abstract does not fully explain the meaning of the paper. The suggestion to the authors is to be specific and precise in the abstract, to provide insight into the analyzes performed, as well as specific results.

(3)In abstract, there should not cite any references. So, [28-30] should be removed.

(4)In Section 1, please improve the introduction by reviewing the latest technology in the paper, reflecting the innovation and research significance of the algorithm in this paper.

(5)In Section 1, It is recommended to use past or present perfect tenses in the literature review.

(6)In Section 3, the applicable scenarios of this research should be accurately described, and the corresponding scientific methods should be introduced for specific problems.

(7)In Section 3.1, it may be necessary to describe the image alignment process in detail.

(8)In Section 4.1.1, the description of the results of the image sequence alignment operation processing should be supplemented, so as to highlight the innovation of the algorithm in this paper.

(9)In Section 4.2, as we know, these datasets come from reference [18]. Please explain the rationale for choosing this data set.

(10)In Section 6. it is suggested that this section be improved. Focus more on what is novel about your research; describe the science and significance of the chosen approach.

(11)References should be updated appropriately to better emphasize the advanced nature of this study.

(12)Figures in this paper are not vector images and are not clear to read. They should be updated.

(13)Layouts of some tables are bad and should be optimized, such as Table 2.

(14)Algorithms should have their special titles, such as Algorithm 1, its title is blank.

Author Response

Dear  reviewer:

      First of all, thank you very much for reviewing my manuscript in your busy schedule. Your comments and suggestions are very important to our work. I have revised my manuscript article by article according to your comments, and you can review the specific revisions in the revised manuscript (marked part).
      Secondly, for your comments (1, 9), I have uploaded additional attachments, you can view them separately.
      Finally, thank you very much for your excellent review work. I look forward to hearing from you if you have any questions.

Reviewer 2 Report

In order to solve the accuracy and computational efficiency of image restoration in air-water interface imaging, the author proposes an image reconstruction method that combines lucky region fusion and image registration algorithm to eliminate the shadow problem in the fused image, The JADE and LBFGS algorithms are integrated as an optimization method to speed up the image registration process. Compared with other image registration methods, the proposed method performs better in restoration accuracy, visual effect and computational efficiency.

1.Major comments

       The paper mentioned that the premise of applying the lucky block search strategy is that the corresponding regions of different frames are spatially invariant or approximate, that is, the spatial invariance should be considered. In the original text of the article, the author has little description of this, and it is necessary to highlight how the proposed method considers the premise of spatial invariance.

2.Minor comments

    (1)Supplement the text description of the reconstructed frame picture in the paper

(2)There are few references in this paper in the past three years, most of which are old and cannot highlight the advanced nature of the research so it needs to be supplemented reasonably.

Author Response

Dear reviewer:

     First of all, thank you very much for reviewing my manuscript in your busy schedule. Your review comments and suggestions are very important to our work. For your review comments, we will provide specific explanations in the following.

  • The process of how to eliminate spatial variability between different images in the same video sequence is supplemented in Section 3.1, which you can see from the revised version of the manuscript.
  • For the specific working principle of Figure 1, I have also supplemented it in Section 3.1. The specific content can be seen from the revised version.
  • For reference, I appropriately supplemented and updated the literature related to this study. You can view it from the modified version.

Finally, thank you very much for your excellent review. If you have any questions, I look forward to hearing from you.